# CS-pFedTM: Communication-Efficient and Similarity-based Personalised Federated Learning with Tsetlin Machine

## Abstract

Federated Learning has emerged as a promising framework for privacy-preserving collaborative model training across decentralised data sources. However, data heterogeneity remains a major challenge, adversely affecting both the performance and efficiency of FL systems. To address this issue, we propose CS-pFedTM (Communication-Efficient and Similarity-based Personalised Federated Learning with Tsetlin Machine), a method that jointly incorporates communication-aware resource allocation and heterogeneity-driven personalisation. CS-pFedTM enforces communication budget constraints through adaptive clause allocation and tailors personalisation by using similarity between clients' model parameters as a proxy for data heterogeneity. To further enhance scalability, the proposed framework integrates confidence-based aggregation and class-specific weight masking. Extensive experiments show that CS-pFedTM achieves reductions in communication and runtime costs, with up to $1352\times$ and $206\times$ reductions in upload and download communication respectively, and at least $1.43\times$ improvements in runtime efficiency, while maintaining performance comparable to state-of-the-art personalised FL approaches.

## 1 Introduction

Federated learning (FL) enables clients to train models locally while sharing only model parameters, thereby preserving privacy as sensitive data remain on individual devices (McMahan et al., 2017). Despite its promise, FL continues to face two major challenges: data heterogeneity across clients and communication constraints, both of which limit scalability in real-world systems (Khan et al., 2021). Personalised FL addresses data heterogeneity by combining locally adapted models with shared global knowledge, which requires balancing effective personalisation with communication efficiency (Liang et al., 2020). While existing methods allow adaptation at a coarse, pre-specified level, they often do not provide flexible, fine-grained control that accounts for both communication budgets and the degree of data heterogeneity (Shamsian et al., 2021; Gohari et al., 2024). Furthermore, most approaches rely on deep neural networks (DNNs) (Asad et al., 2023; Lei et al., 2020), which incur high computational and memory costs, limiting their practicality for resource-constrained edge devices (Almanifi et al., 2023; Khan et al., 2021).

To overcome these limitations, we leverage the low-complexity, rule-based Tsetlin Machine (TM), grounded in finite-state automata and game theory, as an efficient alternative to DNNs (Lei et al., 2020; 2021). We propose CS-pFedTM (Communication-Efficient, Similarity-based Personalised Federated Learning with TM), which jointly addresses data heterogeneity and communication efficiency. TM clause parameters are observed to strongly reflect the underlying FL data distribution, motivating heterogeneity-driven personalisation. Clauses are adaptively allocated according to the communication budget, and weight masking is applied to handle locally absent classes, jointly optimising performance and efficiency. Experimental results show that our approach maintains competitive accuracy while significantly improving efficiency.

The main contributions of this work are summarised as follows:

- We propose a novel TM-based FL personalisation scheme in which each client trains both a local and a global model while communicating only the global model. We also incorporate class-specific weight

- masking to improve flexibility by ignoring irrelevant weights, and confidence-based aggregation to enhance performance and stability across heterogeneous clients.

- We demonstrate that similarity between clients' TM parameters reflects underlying system-level data heterogeneity and we exploit this property to adaptively allocate local and global clauses: higher heterogeneity increases the proportion of local clauses to strengthen personalisation, whereas lower heterogeneity shifts the balance towards global clauses to reinforce shared knowledge.

- We develop a budget-constrained allocation mechanism that adjusts the distribution of local and global clauses according to communication limits, enabling efficient and adaptive personalisation under varying communication budgets.

- Through extensive experiments, we show that CS-pFedTM achieves comparable performance to SOTA personalised FL baselines, while improving efficiency by $15.3 - 1352\times$ and $1.08 - 206\times$ in upload and download communication, respectively, and by at least $1.43\times$ in runtime memory and $1.62\times$ in training latency.

## 2 Related Work

In FL, data heterogeneity and communication efficiency are recognised as major challenges (Tan et al., 2023; Asad et al., 2023). Techniques such as quantization (Mao et al., 2022; Reisizadeh et al., 2019; Hönig et al., 2022), sparsification (Qiu et al., 2022; Rothchild et al., 2020), and network pruning (Jiang et al., 2022; Li et al., 2021) have been proposed to reduce communication and computational costs. Alternative architectures, including Binary Neural Networks (BNNs) (Yang et al., 2021) and Tsetlin Machines (TMs) (How et al., 2023), have been employed to further reduce model size and memory usage, thereby improving efficiency.

Beyond efficiency, significant progress has been achieved in addressing data heterogeneity in FL (Imteaj et al., 2022; Tan et al., 2023; Fallah et al., 2020). Approaches such as multi-task learning (Dinh et al., 2020; Smith et al., 2017) couple client-specific models with a global representation, while meta-learning (Fallah et al., 2020; Jiang et al., 2023) enables rapid local adaptation. Clustering (Sattler et al., 2021) has been used to group similar clients, and knowledge distillation (Li & Wang, 2019) transfers knowledge via teacher–student frameworks. Personalisation via latent distribution modelling (Marfoq et al., 2021; Mclaughlin & Su, 2024) has been employed to explicitly capture data variability, balancing local flexibility with global generalisation.

A complementary line of work has sought to simultaneously address personalisation and communication efficiency. Parameter decoupling methods, such as LG-FedAvg, FedRep, FedBABU, FedPer, and FedPAC (Liang et al., 2020; Collins et al., 2023; Oh et al., 2022; Arivazhagan et al., 2019; Xu et al., 2023), separate client-specific and global components but remain coarse-grained and fixed. FedSelect (Tamirisa et al., 2024), inspired by the Lottery Ticket Hypothesis, discovers fine-grained subnetworks via parameter masks based on importance scores and communicates only the unmasked parameters. Similarly, sparsification-based personalisation methods such as DisPFL, a decentralised FL approach, prune dynamically to exchange only active weights between clients (Dai et al., 2022), and SpaFL communicates only trainable thresholds, reducing communication by two orders of magnitude (Kim et al., 2024) . While effective, these approaches continue to impose structural constraints and do not adaptively allocate shared versus local parameters according to client heterogeneity.

TM-based FL methods, such as FedTM (How et al., 2023) and FedTMOS (How et al., 2025), have been employed. However, FedTM does not address data heterogeneity, whereas FedTMOS targets one-shot FL. The more recent Tsetlin-Personalized Federated Learning (TPFL) (Gohari et al., 2024) introduces personalisation via confidence-based clustering, whereby clients within clusters that share similar class-wise confidence profiles are aggregated. Although TPFL incorporates a degree of personalisation, adaptive adjustment of the balance between local and global TM components is not performed, nor are communication constraints taken into account during the personalisation process.

## 3 Background

### 3.1 Tsetlin Machine

Tsetlin Machine (TM) is a machine learning algorithm in which propositional logic is employed to capture frequent patterns. It is operated using Tsetlin Automata (TA) arranged in teams, with discriminative conjunctive clauses being constructed and a majority voting mechanism being utilised for final classification (Granmo, 2021).

#### 3.1.1 Tsetlin Machine Structure

The structure of a TM is based on two-action TAs and is grounded in reinforcement learning principles.

Consider an input vector of $o$ propositional variables, $\mathbf{x} = \{x_1, \ldots, x_o\} \in \{0, 1\}^o$. Together with their negated counterparts, $\{\neg x_1, \ldots, \neg x_o\}$, these variables form the literal set

$$L = \{l_1, \ldots, l_{2o}\} = \{x_1, \ldots, x_o, \neg x_1, \ldots, \neg x_o\}.$$

The structure of each conjunctive clause $C_j(\mathbf{x})$, indexed by $j$, is determined by selecting literals through a team of $2o$ TAs. Each clause is formed by applying the logical AND operation to a subset $L_j \subseteq L$:

$$C_j(\mathbf{x}) = \bigwedge_{l_k \in L_j} l_k.$$

For a TM with $n$ clauses, a total of $2o \cdot n$ TAs are employed. Each TA decides whether the associated literal is to be included in or excluded from the conjunctive clause, thereby collectively defining the clause structure.

#### 3.1.2 Tsetlin Machine Learning Mechanism

Learning in the TM is initiated by converting the training data into boolean form, thereby enabling the construction of conjunctive clauses from literals, which consist of input variables and their negations. For a TM with $n$ clauses, $n/2$ positive clauses are designated to identify class $y = 1$, and $n/2$ negative clauses are designated to identify class $y = 0$. Training is conducted online, processing one example $(\mathbf{x}, y)$ at a time.

Given an example $(\mathbf{x}, y)$, the TAs are adjusted through two types of feedback, which determine whether literals are to be included in clauses that contribute to a particular class. Type I Feedback reinforces clauses associated with the correct class, increasing the likelihood of outputting 1, whereas Type II Feedback suppresses clauses that would otherwise produce false positives. Feedback is applied to a randomly selected subset of clauses, controlled by the hyperparameter $T$, such that the sum

$$s(\mathbf{x}) = \sum_{j=1}^{n/2} C_j^+(\mathbf{x}) - \sum_{j=n/2+1}^{n} C_j^-(\mathbf{x})$$

approaches $-T$ for $y = 0$ and $T$ for $y = 1$.

and the target:

$$p_y(\mathbf{x}) = \begin{cases} \frac{T + s(\mathbf{x})}{2T}, & \text{if } y = 0 \\ \frac{T - s(\mathbf{x})}{2T}, & \text{if } y = 1 \end{cases} \tag{1}$$

The randomized clause selection ensures a diverse distribution of feedback, preventing clustering on specific patterns and promoting recognition of various sub-patterns. Consequently, the TM progressively refines clause evaluations over successive training iterations, adapting to class-specific objectives and enhancing pattern recognition capabilities.

**Weighted Tsetlin Machine:** In the weighted TM, positive real-valued weights are assigned to individual clauses, enabling a more concise representation of the clause collection. By adjusting these weights, the

influence of particular clauses on the final decision can be controlled, resulting in a real-valued sum within the TM (Phoulady et al., 2020):

$$s(\mathbf{x}) = \sum_{j=1}^{n/2} w_j^+ C_j^+(\mathbf{x}) - \sum_{j=n/2+1}^{n} w_j^- C_j^-(\mathbf{x}).$$

**Multi-Class Tsetlin Machine:** For classification, a unit step function is applied to the sum $s(\mathbf{x})$, producing $y = 0$ if the sum is negative and $y = 1$ otherwise. In multi-class scenarios, each class $m \in \{1, \ldots, M\}$ is associated with a dedicated TA team. For an example $(\mathbf{x}, y = k)$, the TA teams corresponding to class $k$ are trained with $y = 1$, while a randomly selected class $l \neq k$ is trained with $y = 0$. The predicted class is determined by selecting the class with the largest weighted sum:

$$\hat{y} = \arg \max_{m=1,\ldots,M} s^m(\mathbf{x}) = \arg \max_{m=1,\ldots,M} \left( \sum_{j=1}^{n/2} w_j^{+,m} C_j^{+,m}(\mathbf{x}) - \sum_{j=n/2+1}^{n} w_j^{-,m} C_j^{-,m}(\mathbf{x}) \right). \tag{2}$$

**Convolutional Tsetlin Machine (CTM):** Inspired by convolutional operations in deep neural networks, the CTM utilises filters with spatial dimensions $W \times W$ and $Z$ binary layers. An image of size $X \times Y$ with $Z$ binary layers is represented as an input vector $\mathbf{x} = \{x_k \mid k \in \{0,1\}^{X \times Y \times Z}\}$. Clauses act as filters, each comprising $X \times Y \times Z \times 2$ literals (Granmo et al., 2019). For an image containing $B$ patches, each clause outputs $B$ values, one per patch, which are then consolidated via a logical OR:

$$c_j = \bigvee_{b=1}^{B} c_j^b.$$

During training, a single patch is randomly selected from those contributing to a clause evaluating to 1, and feedback is applied based on this patch, following the standard Type I and Type II feedback rules.

**TM Composites:** TM Composites (Granmo, 2023) involve multiple independently trained TMs. Rather than using the $\arg \max$ of a single sum, class sums $s_t^m(\mathbf{x})$ are computed for each TM $t \in \{1, \ldots, r\}$ and normalised by

$$\alpha_t = \max_m(s_t^m(\mathbf{x})) - \min_m(s_t^m(\mathbf{x})).$$

The final predicted class is obtained by

$$\hat{y} = \arg \max_m \left( \sum_{t=1}^{r} \frac{1}{\alpha_t} s_t^m(\mathbf{x}) \right), \tag{3}$$

ensuring that contributions from all constituent TMs are appropriately weighted and combined.

## 3.2 FedTM: Federated Learning with Tsetlin Machine

FedTM represents the first FL framework in which the TM is employed to jointly optimise communication efficiency and memory usage. Unlike FL frameworks based on DNNs, where weight aggregation typically consists of a straightforward weighted averaging of integer weights, a distinctive two-step aggregation scheme is adopted in FedTM (How et al., 2023), reflecting the unique structure of the TM described in Section 3.1.

In the first step, the **TopK** algorithm is used for bit-level aggregation of the TA states. For each class, $K$ clients with the highest number of local samples are aggregated. In the second step, the **AverageCW** method is applied to compute the average of the integer clause weights, weighted according to the total sample size of each client's local data. Previous and current parameters are aggregated as a weighted sum, with the current parameters scaled by $\delta$ and the previous parameters by $(1 - \delta)$, preserving information from integer components of the TM.

While this approach performs well under IID conditions, it does not adequately address the challenges posed by data heterogeneity, similar to the limitations observed in FedAvg (Arivazhagan et al., 2019). The full algorithm is provided in Appendix A.2.1.

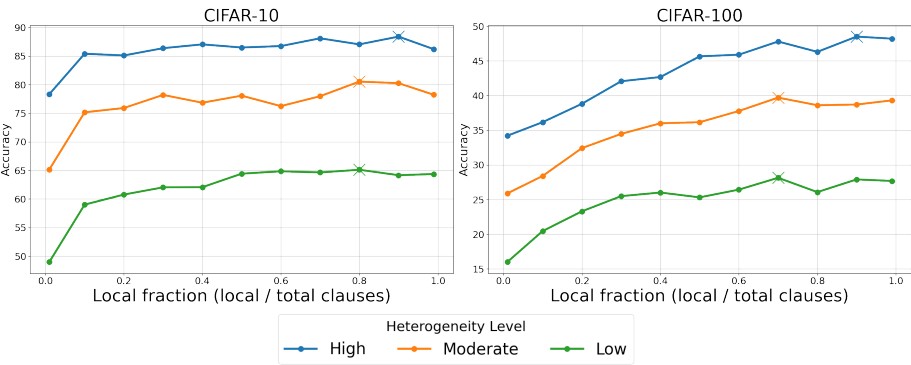

Figure 1: Effect of the local clause fraction on performance. The peak performance is observed to shift towards higher local fractions as data heterogeneity increases.

## 4 Methodology

Before the full method is introduced, the personalisation scheme underlying CS-pFedTM is first presented. This scheme addresses limitations in FedTM for handling data heterogeneity (How et al., 2023). Building on this foundation, CS-pFedTM adapts global and local clause allocations according to system-level data heterogeneity and communication constraints, thereby achieving a balanced trade-off between personalisation and efficiency.

### 4.1 Personalisation

The proposed personalisation strategy enhances the adaptability of each client's model to its local data while still leveraging shared global knowledge. Each client maintains two independent TMs: a local TM, trained exclusively on its own data to capture client-specific patterns, and a global TM, trained on the same local data but whose parameters are uploaded to the server for aggregation. During each communication round, only the global TM parameters of participating clients are sent to the server. After aggregation, the server broadcasts the updated global TM to all clients.

At the client side, the outputs of the local and global TMs are combined using Equation 3. Further personalisation is achieved through class-specific weight masking, where weights corresponding to classes not observed locally are set to zero, enabling rapid adaptation if such classes are encountered in future rounds. These mechanisms collectively support effective personalisation under heterogeneous data distributions.

### 4.2 Effect of Data Heterogeneity on Personalisation

Although the described personalisation framework enables clients to adapt effectively to heterogeneous data, the allocation of clauses between local and global components has a direct influence on both performance and efficiency. As illustrated in Figure 1, accuracy declines when the local fraction is too small, as insufficient clauses are available to support effective personalisation, while allocating almost all clauses locally also leads to suboptimal performance due to reduced sharing of global structure. Under data heterogeneity, a larger proportion of local clauses improves clients' performance, whereas lower data heterogeneity favours global clauses to reinforce shared knowledge. At the same time, communication constraints limit the amount of information that can be exchanged during each round.

The central challenge, therefore, is to determine an allocation of local and global clauses that maximises performance while adhering to communication budgets, without requiring clients to disclose explicit metadata about their data distributions. These observations indicate that no fixed allocation is optimal across all system-level data heterogeneity regimes, motivating the development of CS-pFedTM, a communication-efficient personalisation framework in which the local–global clause ratio is adaptively adjusted based on data heterogeneity.

### 4.3 Exploring the Connection Between Trained Parameters and Distribution Distances

TMs are sensitive to data distributions due to their stochastic clause-update mechanism. Clauses are selected to receive feedback based on Equation 1, reinforcing those that aid correct classification and suppressing those that cause misclassification. Thus, learned clause sets implicitly encode the statistical characteristics of the training data (Tarasyuk et al., 2026).

Each client produces TM parameters shaped by its local data distribution. In FL, parameter similarity inversely reflects data heterogeneity: lower similarity for highly heterogeneous data, higher similarity for aligned distributions.

Let $W(q_A, q_B)$ denote the Wasserstein distance between two data distributions, and $\mathcal{J}(S_A, S_B)$ the Jaccard similarity between their trained TM parameters, quantifying the overlap of active clauses between models trained on different distributions.

**Definition 1 (1-Wasserstein Distance (Kolouri et al., 2017))** *The 1-Wasserstein distance between two distributions $q_1$ and $q_2$ over a metric space $Z$ is*

$$W(q_1, q_2) := \inf_{q \in Q(q_1, q_2)} \int_{Z \times Z} d(z_1, z_2) \, dq(z_1, z_2),$$

*where $d(\cdot, \cdot)$ is a distance function and $Q(q_1, q_2)$ denotes the set of couplings with marginals $q_1$ and $q_2$.*

**Lemma 1 (Distributional Dissimilarity)** *Let $q_1, q_2, q_2'$ be data distributions. If $W(q_1, q_2') > W(q_1, q_2)$, then $q_2'$ is more dissimilar to $q_1$ than $q_2$ is.*

By the definition of the 1-Wasserstein distance and the principle of optimal transport (Kolouri et al., 2017), a larger value indicates that, on average, it is "harder" to transport samples from $q_1$ to $q_2$. Hence, if $W(q_1, q_2') > W(q_1, q_2)$, the distribution $q_2'$ is more dissimilar to $q_1$ than $q_2$ is. Intuitively, samples from $q_2'$ are less likely to resemble samples from $q_1$ compared to samples from $q_2$.

Next, the similarity of parameters is defined by:

**Definition 2 (Jaccard Similarity (da F. Costa, 2021))** *The Jaccard similarity between two sets of binary vectors $S_A$ and $S_B$ is the size of the intersection divided by the size of the union of the sets:*

$$\mathcal{J}(S_A, S_B) = \frac{|S_A \cap S_B|}{|S_A \cup S_B|}.$$

To compare the states between two sets of clauses $A$ and $B$:

- $|S_A \cap S_B|$: represents the number of clauses that are active in both sets, (clauses that has at least one Include action)

- $|S_A \cup S_B|$: represents the number of clauses that contain literals in either $S_A$ or $S_B$.

The Jaccard similarity between two states, $S_A$ and $S_B$, therefore measures the degree of overlap in active clauses between the two states. Higher values indicate that the same clauses has at least an include action in both states, reflecting the similarity in how feedback has shaped the clauses during training.

Empirical results in Figure 2(a) show a strong positive correlation between the Jaccard similarity of clients' learned parameters, $\mathcal{J}(\text{clients})$, and the similarity between the clients' parameters and those trained on the overall data distribution, $\mathcal{J}(\text{true})$. This finding suggests that $\mathcal{J}(\text{clients})$ can be a reliable indicator for $\mathcal{J}(\text{true})$. When client parameters exhibit high mutual agreement, they are also closely aligned with parameters learned directly from the overall distribution.

Overall data heterogeneity can be quantified using $W(\text{true})$, defined as the average Wasserstein distance between the clients' data distributions and the overall distribution. In FL, neither $\mathcal{J}(\text{true})$ nor the overall

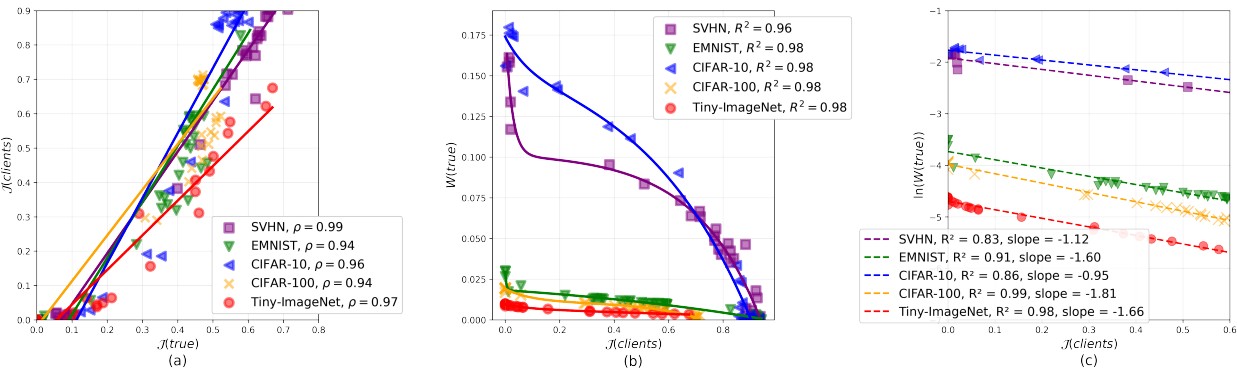

Figure 2: (a) Strong positive correlation and consistent trend between $\mathcal{J}(\text{true})$ and $\mathcal{J}(\text{clients})$. (b) The relationship between $W(\text{true})$ and $\mathcal{J}(\text{clients})$ is well captured by a bi-exponential decay model, achieving an average of $R^2 = 0.98$ across all datasets. c) Semi-logarithmic plot of $\ln(W(\text{true}))$ versus $\mathcal{J}(\text{clients})$, restricted to the high-heterogeneity regime ($\mathcal{J}(\text{clients}) \leq 0.6$), where linear regression is fitted within this range. The linear trend across all datasets ($R^2 \geq 0.83$, $\beta < 0$) suggests that a decreasing exponential is a reasonable approximation in this regime.

distribution is accessible, as data remains decentralised. Although $\mathcal{J}(\text{true})$ and $W(\text{true})$ capture related aspects of proximity to the overall distribution, both are unavailable without centralised access to all data. The practical question is therefore whether $\mathcal{J}(\text{clients})$ can serve as a reliable proxy for $W(\text{true})$.

This relationship is demonstrated in Figure 2(b), where a decreasing bi-exponential model captures the relationship between $\mathcal{J}(\text{clients})$ and $W(\text{true})$, achieving an average goodness-of-fit of $R^2 = 0.98$ across all datasets. The exponential form is preferred over linear alternatives, as it captures the expected saturation behaviour and avoids unrealistic extrapolation. Furthermore, as interest lies primarily in the high-heterogeneity regime (where $W(\text{true})$ is large and $\mathcal{J}(\text{clients})$ is small), the semi-logarithmic plot in Figure 2(c) of $\ln(W(\text{true}))$ versus $\mathcal{J}(\text{clients})$ exhibits an approximately linear trend in this regime ($R^2 \geq 0.83$, slope $< 0$), consistent with an underlying exponential decay. Notably, this log-linear fit improves with task complexity, reaching $R^2 = 0.98$ and $R^2 = 0.99$ for Tiny-ImageNet (200 classes) and CIFAR-100 (100 classes) respectively, suggesting that the exponential character becomes more pronounced as the number of classes increases. A decreasing exponential functional form is therefore a well-motivated approximation for this critical region, providing a bounded and interpretable mapping well-suited for guiding similarity-driven clause allocation without requiring access to client metadata or the overall distribution.

### 4.4 Communication-Aware Clause Allocation

A communication-aware allocation mechanism is introduced to address personalisation under communication constraints. Given a communication budget $\tau$, which specifies the maximum number of megabytes that each client is permitted to communicate per round, the reference TM is first used to estimate the per-clause communication footprint, including both clause weights and TA states. This estimation enables the abstract budget $\tau$ to be translated into a concrete limit on the number of clauses that can be shared globally without exceeding the permitted communication cost.

On the basis of this limit, `min_frac` is computed, representing the minimum fraction of clauses that must remain local. This ensures that each client retains a sufficient number of locally trained clauses while remaining within the communication budget, yet still benefits from global aggregation. By enforcing this budget-driven lower bound, the mechanism prevents infeasible allocations, preserves fairness across heterogeneous clients, and establishes a stable foundation for the similarity-driven personalisation step, which subsequently allocates clauses adaptively according to data heterogeneity. The effect of varying $\tau$ on model performance is reported in Appendix A.3.1, where performance across a range of values demonstrate that CS-pFedTM remains stable and robust to the choice of communication budget.

### 4.4.1 Similarity-Driven Personalisation

Within the communication limit established by the budget, clause allocation is further adapted on the basis of data heterogeneity. As illustrated in Figure 1, higher heterogeneity ($W(\text{true})$) is associated with improved performance when a larger fraction of clauses is kept local. Because $W(\text{true})$ is not observable in FL, it is approximated using $\mathcal{J}(\text{clients})$, the average similarity between clients' trained TM parameters. Figure 2 demonstrates a strong inverse relationship between $\mathcal{J}(\text{clients})$ and $W(\text{true})$, with client parameters becoming less similar as their data distributions deviate further from the system-wide distribution.

To translate this relationship into clause allocation, a scalar `local_frac` is introduced, representing the fraction of the total clause budget assigned to local training. The remaining fraction $1 - \texttt{local\_frac}$ is allocated to global clauses, which are shared and aggregated across clients. A higher value of `local_frac` therefore increases personalisation, while a lower value promotes global consensus.

While a linear model achieves a comparable fit ($R^2 = 0.93$), the exponential form is preferred for the allocation function on structural grounds rather than fit quality alone. A linear mapping lacks natural bounds, requiring artificial clipping to ensure $\texttt{local\_frac} \in (0, 1]$ across all possible values of $\mathcal{J}(\text{clients})$. The exponential formulation $\exp(-c \cdot \mathcal{J}(\text{clients}))$ satisfies these bounds by construction for all inputs. Motivated by the empirically observed exponential character of the $\mathcal{J}(\text{clients})$–$W(\text{true})$ relationship in the high-heterogeneity regime (Figure 2(c)), a decreasing exponential is adopted to drive clause allocation, where the constant $c = \ln(1/\texttt{min\_frac})$ directly encodes the communication budget constraint analytically rather than through dataset-specific fitting. Specifically, this ensures that

$$\exp\big(-c \cdot \mathcal{J}(\text{clients})\big) \geq \texttt{min\_frac}$$

for all inputs. The resulting local fraction is therefore defined as:

$$\texttt{local\_frac} = \exp\big(-c \cdot \mathcal{J}(\text{clients})\big).$$

The corresponding numbers of local and global clauses are then given by:

$$n_{\text{local}} = \lfloor n_{\text{clauses}} \cdot \texttt{local\_frac} \rfloor, \qquad n_{\text{global}} = n_{\text{clauses}} - n_{\text{local}}.$$

## 4.5 Modified Global Aggregation Scheme

To improve robustness under data heterogeneity, two modifications to the aggregation strategy used in FedTM are introduced: a confidence-based state aggregation scheme, referred to as $\textbf{TopK}_{\textbf{Conf}}$, and a clause weight aggregation scheme, denoted as $\textbf{AverageCW}_{\textbf{CA}}$.

In TMs, prolonged training over multiple rounds can cause clause votes to saturate towards the feedback threshold $T$, limiting adaptability and incorporating parameters from previous rounds via $\textbf{AverageCW}$ has also been observed to propagate errors under highly non-IID data (How et al., 2023). In the personalised setting, where class-specific masking is applied, only the current-round global parameters are relevant for each client. Therefore, aggregation has been simplified, with the scaling factor $\delta$ applied exclusively to the current-round global parameters:

$$\mathbf{W}_t[m] \leftarrow \delta \left( \frac{1}{|D|} \sum_{j=1}^{J} |D_j| \, \mathbf{W}_t^j[m] \right) \tag{4}$$

This modification, $\textbf{AverageCW}_{\textbf{CA}}$, stabilises aggregation, prevents the propagation of errors across rounds, and aligns with the personalised objective of CS-pFedTM.

To protect client data, a confidence-based aggregation strategy, $\textbf{TopK}_{\textbf{Conf}}$, is employed instead of sample-based $\textbf{TopK}$ aggregation, which could risk meta-data leakage. For each class, the top $K$ clients with the highest model confidence are selected to update the global model. Using actual model confidence is preferred over local sample counts, which can bias selection towards clients with larger datasets irrespective of the quality of their models. For client $j$, confidence is computed as:

$$\text{Confidence}_j[m] \leftarrow \frac{1}{r} \sum_{t=1}^{r} \frac{1}{|D_j|} \sum_{\mathbf{x} \in D_j} \frac{s_t^m(\mathbf{x}) - \min_{\mathbf{x}' \in D_j} s_t(\mathbf{x}')}{\max_{\mathbf{x}' \in D_j} s_t(\mathbf{x}') - \min_{\mathbf{x}' \in D_j} s_t(\mathbf{x}')} \tag{5}$$

where $|D_j|$ is the number of local samples, $[m]$ denotes the class, and $r$ is the number of TMs in the ensemble. Since each model outputs votes, this serves as a confidence score that enables the server to reliably identify the most confident clients without sharing raw samples or other meta-data, thereby improving the quality of aggregation while preserving fairness across clients. The impact of the modified aggregation scheme on convergence behaviour and performance is evaluated in Appendix A.3.2.

### 4.6 Algorithm Overview

CS-pFedTM begins with a reference round in which clients train a small reference TM and upload their parameters to the server. These reference parameters serve two purposes: estimating communication costs per clause for download budgeting, and computing pairwise client parameter similarity to estimate system-level data heterogeneity. Based on this similarity, the server allocates local and global clauses: higher heterogeneity increases local clauses to strengthen personalisation, whereas lower heterogeneity favours global clauses for knowledge sharing.

In subsequent rounds, clients are randomly sampled for participation. Only the model states of the top-confidence clients are uploaded, determined by their local output confidence. After aggregation, the server broadcasts the updated global model to all clients. Algorithm 1 summarises the full approach.

---

**Algorithm 1 CS-pFedTM: Communication-Efficient and Similarity-Based Personalised FL with TM**

---

**Input:** Total number of clients $N_c$, total communication rounds $T$, number of clauses per client $n_{\text{clauses}}$, communication budget $\tau$

  **for** round $t = 0, 1, \ldots, T$ **do**

    The server randomly samples $N_t$ clients, $\mathcal{C}_t$

    **if** $t == 0$ **then**

      Clients train a small reference TM and upload their state parameters

      `min_frac` $\leftarrow$ **compute_min_frac**

      $JS_{\text{clients}} \leftarrow$ **compute_client_similarity**

      `local_frac` $\leftarrow \exp\left(-\ln(1/\texttt{min\_frac}) \cdot JS_{\text{clients}}\right)$

      Local and global clauses are assigned as:

$$n_{\text{local}} = \lfloor n_{\text{clauses}} \cdot \texttt{local\_frac} \rfloor, \quad n_{\text{global}} = n_{\text{clauses}} - n_{\text{local}}$$

    **for** each client $n \in \mathcal{C}_t$ **do**

      The client trains a local model $L^n$ and a global model $G^n$

      $L^n, G^n \leftarrow$ **mask_weights**$(L^n)$, **mask_weights**$(G^n)$

      The client uploads global parameters $G^n$ to the server

    $G_t \leftarrow$ **aggregate_global_models**

    Clients download the global TM, $G_t$

  **return** Personalised TMs for each client: $TM^n \in \{G_t, L^n\}$, combined using Equation 3.

---

## 5 Experiments

**Benchmark Datasets:** We performed experiments on five image datasets commonly featured in the FL literature: SVHN (Netzer et al., 2011), EMNIST (Cohen et al., 2017), CIFAR-10, CIFAR-100 (Krizhevsky, 2009) and Tiny-ImageNet (Le & Yang, 2015).

**Baseline Methods:** To ensure a fair comparison, several parameter–decoupling personalisation approaches are evaluated alongside CS-pFedTM. FedAvg is used as the standard FL benchmark (McMahan et al., 2017),

Table 1: Performance under data heterogeneity with upload/download communication costs (CC) per round in MB for SVHN, EMNIST, and Tiny-ImageNet.

| | SVHN | | | | EMNIST | | | | Tiny-ImageNet | | | |
|---|---|---|---|---|---|---|---|---|---|---|---|---|
| | $Dir(0.05)$ | | $Dir(0.1)$ | | $Dir(0.05)$ | | $Dir(0.1)$ | | $Dir(0.05)$ | | $Dir(0.1)$ | |
| | Acc | CC | Acc | CC | Acc | CC | Acc | CC | Acc | CC | Acc | CC |
| FedAvg | 34.67±12.96 | 13/43 | 58.43±3.60 | 13/43 | 73.51±4.25 | 15/50 | 77.68±3.33 | 15/50 | 1.69±0.28 | 16/53 | 1.66±0.25 | 16/53 |
| FedAvg++ | 79.66±1.89 | " | 69.90±2.94 | " | 73.00±1.06 | " | 75.18±0.95 | " | 18.87±1.87 | " | 18.76±2.17 | " |
| pFedFDA | 80.07±2.59 | " | 68.95±3.53 | " | 95.73±0.14 | 14/46 | 94.26±0.20 | 14/46 | 28.29±0.73 | 13/43 | 23.46±0.63 | 13/43 |
| FedPAC | 82.20±1.89 | " | 83.85±1.88 | " | 95.99±0.64 | " | 94.55±0.40 | " | 28.95±0.64 | " | 22.68±0.61 | " |
| FedRep | 80.04±2.47 | " | 80.34±2.25 | " | 78.41±0.42 | " | 78.55±0.55 | " | 27.70±0.70 | " | 20.60±0.43 | " |
| FedPer | 81.41±1.85 | " | 76.68±1.35 | " | 94.72±0.87 | " | 93.07±0.62 | " | 28.67±0.78 | " | 23.52±0.89 | " |
| LG-FedAvg | 83.70±1.56 | 0.67/1.0 | 78.37±1.06 | 0.67/1.0 | 76.34±0.84 | 4.2/6.4 | 75.58±0.34 | 4.2/6.4 | 23.24±1.28 | 13/20 | 15.21±0.50 | 13/20 |
| FedSelect | 76.94±2.10 | 8.38/28.6 | 66.87±2.04 | 8.32/28.5 | 94.83±0.43 | 9.65/32.8 | 92.37±0.39 | 9.59/32.7 | 27.87±0.86 | 10.3/34.9 | 21.12±0.89 | 10.3/35 |
| TPFL | 89.27±2.03 | 0.19/0.63 | 83.60±0.44 | 0.19/0.63 | 94.13±0.75 | 0.20/0.67 | 90.99±1.02 | 0.20/0.67 | 28.82±0.43 | 0.076/0.25 | 23.03±0.55 | 0.076/0.25 |
| FedTM | 32.80±3.94 | 0.33/12 | 27.65±1.90 | 0.33/12 | 40.50±1.82 | 1.4/48 | 39.18±0.59 | 1.4/48 | 2.04±0.60 | 1.4/53 | 1.65±0.32 | 1.4/53 |
| CS-pFedTM | 89.42±0.60 | 0.0058/0.12 | 83.90±1.01 | 0.0077/0.16 | 94.36±0.60 | 0.012/0.25 | 91.33±0.46 | 0.024/0.50 | 29.00±0.61 | 0.04/0.88 | 23.66±0.81 | 0.04/0.88 |

while FedAvg++ incorporates local fine-tuning (Jiang et al., 2023). pFedFDA addresses the bias–variance trade-off through generative classifiers and feature distribution adaptation (Mclaughlin & Su, 2024). FedPAC aligns local and global feature representations via a regularisation term (Xu et al., 2023). FedRep and FedPer communicate only the base layers, with classifier heads or the full model retrained locally for personalisation (Collins et al., 2023; Arivazhagan et al., 2019). LG-FedAvg transmits only the global classifier and linearly combines local and global layers (Liang et al., 2020) and FedSelect personalises subnetworks through selective masking (Tamirisa et al., 2024). TM-based FL baselines are also considered. FedTM serves as a baseline TM-based FL approach, which does not explicitly address data heterogeneity (How et al., 2023), whereas TPFL addresses heterogeneity via confidence-based client clustering (Gohari et al., 2024).

**FL Configuration:** Following standard practice (Hsu et al., 2019; Jiang et al., 2023; Mclaughlin & Su, 2024), data heterogeneity is simulated using a Dirichlet parameter $\alpha \in \{0.1, 0.05\}$. Across 100 clients, a participation rate of 0.3 is used: only the selected clients train on their local models and upload their updates to the server, while the server broadcasts the aggregated model to all clients. Unless stated otherwise, each client trains for one local epoch per round, and the highest average personalised accuracy of all clients after 100 rounds is reported, averaged over five runs. Communication cost is measured as the total number of parameters uploaded and downloaded per round. Personalised accuracy is averaged across all clients, with full-participation results also reported for completeness.

**Model Configuration:** A 2-layer CNN (Xu et al., 2023) is used for the image datasets, with training performed using a batch size of 128 (Liang et al., 2020). Parameter tuning is carried out for all baseline models to ensure a fair comparison. In particular, when a learning rate is not specified in the original paper, values in the range $[0.01, 0.05, 0.1]$ are tuned. FedTM and CS-pFedTM employ CTMs, whereas TPFL uses a Coalesced TM (Glimsdal & Granmo, 2021). To ensure a fair comparison, the static model size of CTM and CoTM is matched. Since TPFL's CoTM reduces static model size by coalescing clauses across classes, this matching requires CoTM to employ a larger number of clauses. The download budget, $\tau$, for CS-pFedTM is set to match that of the most download-efficient baseline, providing comparable communication conditions.

## 5.1 Performance

CS-pFedTM achieves accuracy comparable to SOTA personalised FL methods across all heterogeneous settings. It consistently outperforms TM-based FL baselines, namely FedTM and TPFL, with average accuracy improvements of 48.8% and 0.65%, respectively. However, with the same number of clauses, CS-pFedTM outperforms TPFL by 4.58% on average as reported in Table 4. In addition, CS-pFedTM is evaluated against sparsification-based personalisation methods. Since DisPFL operates in a decentralised setting and SpaFL relies on larger CNN architectures to enable pruning, these comparisons are reported separately in Appendix A.3.7, where CS-pFedTM demonstrates improved efficiency and performance.

Table 2: Performance under data heterogeneity with upload/download communication costs (CC) per round in MB for CIFAR-10 and CIFAR-100.

| | CIFAR-10 | | | | CIFAR-100 | | | |
| | $Dir(0.05)$ | | $Dir(0.1)$ | | $Dir(0.05)$ | | $Dir(0.1)$ | |
| | Acc | CC | Acc | CC | Acc | CC | Acc | CC |
|---|---|---|---|---|---|---|---|---|
| FedAvg | 31.86±1.04 | 13/43 | 33.34±0.73 | 13/43 | 6.47±0.64 | 14/48 | 6.70±0.22 | 14/48 |
| FedAvg++ | 79.51±2.63 | " | 68.46±2.21 | " | 33.02±0.60 | " | 22.17±0.93 | " |
| pFedFDA | 84.91±1.63 | " | 77.00±0.65 | " | 47.19±1.00 | " | 38.56±1.36 | " |
| FedPAC | 85.45±1.01 | " | 79.34±0.60 | " | 45.71±0.56 | " | 37.35±0.59 | " |
| FedRep | 86.58±1.36 | " | 79.52±1.07 | " | 44.50±1.24 | " | 36.99±1.23 | " |
| FedPer | 84.34±1.59 | " | 76.16±0.96 | " | 44.93±0.91 | " | 37.81±0.72 | " |
| LG-FedAvg | 83.81±1.30 | 0.67/1.0 | 75.57±0.86 | 0.67/1.0 | 37.64±0.91 | 6.7/10 | 28.91±0.57 | 6.7/10 |
| FedSelect | 83.71±0.62 | 8.34/28.5 | 75.84±0.83 | 8.37/28.5 | 41.65±0.69 | 9.24/31.7 | 32.38±0.68 | 9.24/31.6 |
| TPFL | 86.81±1.50 | 0.13/0.43 | 78.83±1.11 | 0.13/0.43 | 46.89±0.85 | 0.12/**0.41** | 37.24±0.36 | 0.12/0.41 |
| FedTM | 15.81±0.81 | 0.37/15 | 13.69±2.75 | 0.37/15 | 2.35±1.41 | 1.3/46 | 2.59±0.19 | 1.3/46 |
| CS-pFedTM | **86.87±0.75** | **0.004**/**0.09** | **79.84±0.86** | **0.006**/**0.13** | **48.29±0.63** | **0.028**/0.44 | **39.42±0.86** | **0.014**/**0.35** |

Table 3: Average memory storage (MS) and run-time memory (RTM) in MB, and training latency (L) in seconds on each client.

| | SVHN | | | EMNIST | | | CIFAR-10 | | | CIFAR-100 | | | Tiny-ImageNet | | |
| | MS | RTM | L | MS | RTM | L | MS | RTM | L | MS | RTM | L | MS | RTM | L |
|---|---|---|---|---|---|---|---|---|---|---|---|---|---|---|---|
| CNN | 0.43 | 101 | 1.48 | 0.50 | 50.6 | 3.82 | 0.43 | 111 | 1.52 | 0.48 | 118 | 1.47 | **0.53** | 144 | 3.23 |
| CoTM | **0.12** | 25.2 | 4.88 | **0.48** | 75.0 | 24.3 | **0.09** | 32.8 | 6.48 | **0.46** | 34.5 | 9.19 | **0.53** | 57.6 | 21.9 |
| CTM | **0.12** | **22.1** | **0.70** | **0.48** | **47.6** | **3.50** | **0.09** | **19.1** | **0.63** | **0.46** | **25.7** | **1.24** | **0.53** | **42.5** | **2.45** |

## 5.2 Communication Costs

Communication costs, especially upload costs, are a major bottleneck in FL, particularly for edge devices with limited bandwidth (Asad et al., 2023). As shown in Tables 1 and 2, CS-pFedTM achieves the lowest overall communication cost among all evaluated methods. This reduction is primarily driven by its design, in which clients upload only the subset of global parameters allocated according to estimated heterogeneity and communication budgets, rather than the full model. In addition, the bit-based CTM representation further reduces communication overhead compared to full-precision CNNs (Lei et al., 2020).

As a result, CS-pFedTM achieves 63.8× and 110× lower upload and download communication costs than FedTM, respectively. On average, CS-pFedTM is 71.8× more upload-efficient and 105× more download-efficient than LG-FedAvg, and 871× and 139× more efficient than FedSelect in terms of upload and download communication. Although FedTM incurs lower upload costs than LG-FedAvg, its download efficiency remains comparatively weaker. While FedPAC achieves higher accuracy than CS-pFedTM on EMNIST, CS-pFedTM remains, on average, 1352× more upload-efficient and 206× more download-efficient. Furthermore, compared with TPFL, which exchanges class-conditioned clause subsets, CS-pFedTM reduces upload and download communication by 15.3× and 2.3×, respectively.

Overall, these results demonstrate that CS-pFedTM provides the most communication-efficient solution among all baselines, making it particularly well suited for bandwidth-constrained FL deployments.

## 5.3 Efficiency Analysis

Run-time memory usage, latency, and model storage were evaluated for all TM variants and CNNs. As shown in Table 3, CTMs and CoTMs are more storage-efficient than CNNs, requiring 2.29× less storage on average, while CTMs further achieve 3.88× lower run-time memory usage.

Table 4: Performance of TPFL and CS-pFedTM under data heterogeneity, together with upload and download communication costs (CC) per round in MB

| Dataset | Method | $Dir(0.05)$ | | $Dir(0.1)$ | |
|---|---|---|---|---|---|
| | | Acc | CC | Acc | CC |
| SVHN | TPFL | 86.45±0.56 | 0.036/**0.12** | 79.29±1.63 | 0.036/**0.12** |
| | CS-pFedTM | **89.42±0.60** | **0.0058**/0.12 | **83.90±1.01** | **0.0077**/0.16 |
| EMNIST | TPFL | 91.99±0.23 | 0.023/**0.078** | 89.92±1.21 | **0.023/0.078** |
| | CS-pFedTM | **94.36±0.60** | **0.012**/0.2 | **91.33±0.46** | 0.024/0.50 |
| CIFAR-10 | TPFL | 84.72±0.96 | 0.024/**0.08** | 77.53±1.15 | 0.024/**0.08** |
| | CS-pFedTM | **86.87±0.75** | **0.004**/0.09 | **79.84±0.86** | **0.006**/0.13 |
| CIFAR-100 | TPFL | 41.87±0.61 | **0.013/0.042** | 31.90±0.46 | **0.013/0.042** |
| | CS-pFedTM | **48.29±0.63** | 0.028/0.44 | **39.42±0.86** | 0.014/0.35 |
| Tiny-ImageNet | TPFL | 20.76±0.44 | **0.0072/0.024** | 15.82±0.55 | **0.0072/0.024** |
| | CS-pFedTM | **29.00±0.61** | 0.04/0.88 | **23.66±0.81** | 0.04/0.88 |

Table 5: Average memory storage (MS) and run-time memory (RTM) in MB, and training latency (L) in seconds on each client for CTM and CoTM.

| | SVHN | | | EMNIST | | | CIFAR-10 | | | CIFAR-100 | | | Tiny-ImageNet | | |
|---|---|---|---|---|---|---|---|---|---|---|---|---|---|---|---|
| | MS | RTM | L | MS | RTM | L | MS | RTM | L | MS | RTM | L | MS | RTM | L |
| CoTM | **0.03** | 22.5 | 2.92 | **0.06** | 71.5 | 17.1 | **0.02** | 31.8 | 2.44 | **0.05** | 28.2 | 3.36 | **0.05** | 57.3 | 10.6 |
| CTM | 0.12 | **22.1** | **0.70** | 0.48 | **47.6** | **3.50** | 0.09 | **19.1** | **0.63** | 0.46 | **25.7** | **1.24** | 0.53 | **42.5** | **2.45** |

To enable a fair comparison with CS-pFedTM, TPFL, which uses the more compact CoTM representation, was evaluated under a matched static storage budget. Specifically, the number of clauses in CoTM was adjusted so that its static storage matches that of CTM. While this equalises static memory, it does not guarantee identical run-time behaviour. CoTM achieves its storage savings by coalescing clauses across classes, which requires a larger total number of clauses to preserve representational capacity and this increases the computational cost during both inference and training. Despite having matched static storage, CoTM exhibits higher run-time memory usage and longer latency, with increases of up to 1.43× in run-time memory and 8.11× in latency compared to CTM. Furthermore, when CoTM and CTM have the same total number of clauses, CTM still remains more efficient at run-time than CoTM as reported in Table 5. This highlights that CTM offers more favourable run-time efficiency compared to CoTM.

## 5.4 Further Comparison with TPFL

TPFL personalises models via confidence-based clustering, grouping clients with similar class-wise confidence profiles and performing aggregation within each cluster (Gohari et al., 2024). This approach tailors updates to subsets of similar clients and uses the Coalesced Tsetlin Machine (CoTM), a compact TM variant that reduces static memory requirements and has been shown to outperform the standard CTM in centralised settings (Glimsdal & Granmo, 2021).

In the main experimental evaluation (Table 1 and 2), TPFL and CS-pFedTM were compared using the same model size, which required increasing the number of clauses in CoTM for TPFL to match the CTM used in CS-pFedTM. Under this setting, both methods achieve comparable accuracy. However, CS-pFedTM

is significantly more communication- and runtime-efficient, achieving 20.8× lower upload cost, 2.94× lower download cost, 1.4× lower runtime memory, and 9.07× lower latency.

From Table 4, when comparing an equal number of clauses, CS-pFedTM outperforms TPFL by 4.78% in accuracy and is, on average, 2.55× more upload-efficient. TPFL remains 10.7× more download-efficient, while CoTM achieves 7.26× lower static memory usage as shown in Table 5. In contrast, CTM offers better runtime efficiency, with 1.32× lower runtime memory and 3.99× lower latency. While CoTM improves storage and download efficiency due to its compact size, upload costs are typically the primary bottleneck in FL (Asad et al., 2023), which remain a key consideration when selecting the global model.

Importantly, the mechanisms underlying CS-pFedTM, estimating heterogeneity via state similarity and allocating clauses based on this measure, are model-agnostic. For future work, employing a CoTM as the global model could reduce communication costs further, since fewer clause weights would need to be transmitted, while retaining CTM locally would preserve runtime efficiency and client-level personalisation.

## 5.5 Ablation Studies

Ablation experiments were conducted to assess the individual contributions of the two core components of CS-pFedTM: masking and similarity-based personalisation. When applied independently, each component provides only partial improvements under heterogeneous data distributions. Notably, similarity-based personalisation contributes a larger performance gain than masking alone, indicating that adapting models to client heterogeneity is the dominant factor driving accuracy improvements. When combined, the two components yield the highest accuracy and most consistent performance across all datasets. These results demonstrate that the full CS-pFedTM design is required to realise performance gains, with masking and similarity-driven personalisation functioning most effectively when used together.

Table 6: Ablation Studies for CS-pFedTM.

| | Dataset | FedTM | Mask Only | Personalisation Only | CS-pFedTM |
|---|---|---|---|---|---|
| | SVHN | 32.18±4.91 | 47.50±0.93 | 86.31±0.41 | **89.59±0.95** |
| | EMNIST | 40.81±0.63 | 71.58±0.26 | 91.73±0.19 | **94.36±0.40** |
| $Dir(0.05)$ | CIFAR-10 | 15.91±1.14 | 54.58±0.42 | 84.03±0.39 | **86.92±1.18** |
| | CIFAR-100 | 2.63±1.85 | 13.27±0.35 | 45.43±0.39 | **48.09±0.28** |
| | Tiny-ImageNet | 1.67±0.11 | 5.84±0.02 | 26.20±0.29 | **29.24±0.64** |
| | SVHN | 26.67±1.30 | 39.62±0.52 | 79.96±0.65 | **83.67±0.68** |
| | EMNIST | 38.97±0.67 | 71.17±0.09 | 88.16±0.14 | **91.08±0.09** |
| $Dir(0.1)$ | CIFAR-10 | 14.85±3.16 | 46.03±1.33 | 78.94±0.21 | **79.61±1.58** |
| | CIFAR-100 | 2.58±0.21 | 10.23±0.47 | 35.57±0.47 | **38.85±0.35** |
| | Tiny-ImageNet | 1.74±0.19 | 4.26±0.01 | 21.25±0.16 | **23.91±0.32** |

## 6 Conclusions

We presented CS-pFedTM, which introduces a personalised FL framework with TMs that unifies global knowledge sharing with client-specific adaptation through a clause similarity mechanism. The key insight underpinning this approach is that the similarity between TM parameters reflects the heterogeneity of local data distributions. Clients with similar local distributions exhibit similar parameter patterns, whereas heterogeneous clients display more divergent patterns. By exploiting this relationship, clauses are allocated according to parameter similarity, and subjected to communication budgets, ensuring that clients receive a personalised model well-suited to their data while retaining knowledge shared across the global model. CS-pFedTM achieves substantial reductions in resource usage, including up to 1352× lower upload communication, 206× lower download communication, 3.88× lower runtime memory consumption, and 8.11× lower training latency, without compromising performance. These findings highlight that heterogeneity-driven personalisation not only provides an efficient and scalable approach for TM-based FL but also opens avenues for

future research, including dynamic clause adaptation, resource-aware personalisation, and robust handling of client heterogeneity in real-world deployments.

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

## A  Appendix

### A.1  Experimental Details

### A.1.1  Datasets

The proposed approaches were evaluated on the SVHN (Netzer et al., 2011), Extended MNIST (EMNIST) (Cohen et al., 2017), CIFAR-10 and CIFAR-100 (Krizhevsky, 2009), and Tiny-ImageNet (Le & Yang, 2015) datasets. All datasets were downloaded and preprocessed using PyTorch (Paszke et al., 2019).

- **SVHN:** This imbalanced dataset consists of digits captured in natural scenes, presenting a challenging real-world classification problem (Netzer et al., 2011).

- **EMNIST:** The extended version of MNIST contains 814,255 characters across 62 unbalanced classes. Following prior work such as BiFL (Yang et al., 2021), only a subset of the dataset was used for training and testing.

- **CIFAR-10:** A real-world image dataset comprising 10 classes, each with 6,000 images (Krizhevsky, 2009).

- **CIFAR-100:** A real-world image dataset comprising 100 classes, each with 6,000 images (Krizhevsky, 2009).

- **Tiny-ImageNet:** A dataset containing 100,000 images across 200 classes, downsized to $64\times64$ colour images (Le & Yang, 2015).

### A.1.2  Libraries and Machine Configuration

All experiments were conducted on a general-purpose compute node equipped with 32 CPU cores. During training, the average per-client latency and estimated run-time memory usage were recorded. Memory profiling was performed using the `memory-profiler` Python package (Paszke et al., 2019).

### A.1.3  Baseline Models Configuration

In configuring all baseline models, we performed parameter tuning to optimize their performance. specifically, for the learning rate if not defined in the original paper, we explored these values: [0.01, 0.05, 0.1].

### A.1.4  CS-pFedTM Model Configuration

The model used in CS-pFedTM is the CTM, with configurations that vary across datasets. The values of all definable parameters are listed in Table 7. A scaling parameter of $\delta = 0.5$ is used in **AverageCW$_{\mathbf{CA}}$**, which, together with a single local epoch, provides a good balance without excessively scaling the parameters. For **TopK$_{\mathbf{Conf}}$**, $K = 2$ is selected, consistent with FedTM. To satisfy the Boolean input requirements of CTM, dataset-specific preprocessing procedures are applied. For EMNIST, pixel intensities greater than 40 are mapped to 1, and values less than or equal to 40 are mapped to 0. For SVHN, binarisation is performed using adaptive Gaussian thresholding with a window size of 11 and a threshold value of 2 (Granmo et al., 2019). For SpeechCommands-12, a quantile threshold of 0.4 is applied. For CIFAR-10, CIFAR-100, and Tiny-ImageNet, an 8-level colour thermometer encoding is used (Granmo, 2023).

Table 7: CS-pFedTM model configuration.

|  |  | SVHN | EMNIST | CIFAR-10 | CIFAR-100 | Tiny-ImageNet |
|---|---|---|---|---|---|---|
| $Dir(0.05)$ | Number of Local Clauses | 297 | 194 | 198 | 104 | 59 |
|  | Number of Global Clauses | 3 | 1 | 2 | 1 | 1 |
| $Dir(0.1)$ | Number of Local Clauses | 296 | 193 | 197 | 104 | 59 |
|  | Number of Global Clauses | 4 | 2 | 3 | 1 | 1 |
| Feedback Threshold |  | 1000 | 300 | 200 | 180 | 600 |
| Learning Sensitivity |  | 3 | 2 | 3 | 1.5 | 1.2 |
| Patch Dimension |  | (5,5) | (10,10) | (2,2) | (2,2) | (2,2) |

## A.2 Full Algorithm

### A.2.1 FedTM

---
**Algorithm 2 FedTM**
---
The global parameters $\mathbf{W}_0$ and $\mathbf{S}_0$ are initialised with the same TM architecture, and clients inform the server of their dataset sizes $|D_j|$, $j = 1, \ldots, N$.

**for** communication round $t = 1, \ldots, T$ **do**

    Each participating client $J$ trains a TM using the current weights $\mathbf{W}_{t-1}$ on its local dataset $D_j$ for $e$ epochs.

    The trained local parameters are uploaded by the clients.

    The uploaded parameters are aggregated.

    **for** class $m = 1, \ldots, M$ **do**

        $\mathbf{W}_t[m] \leftarrow \mathbf{AverageCW}(m, \delta, t)$

        $\mathbf{S}_t[m] \leftarrow \mathbf{TopK}(m, k, t)$

    The updated global parameters $\mathbf{W}_t$ and $\mathbf{S}_t$ are downloaded by all clients.

---

---
**Algorithm 3 AverageCW**$(m, \delta, t)$
---
A weighted average of local clause weights is computed:

$$\mathbf{W}_t[m] \leftarrow int\left(\frac{1}{|D|} \sum_{j=1}^{J} |D_j| \, \mathbf{W}_t^j[m]\right).$$

**if** $t > 1$ **then**

    **if** $\forall_{j=1}^{J} \mathbf{W}_t^j[m] = 0$ **then**

        The previous weights are retained when class $m$ is not observed:

$$\mathbf{W}_t[m] \leftarrow \mathbf{W}_{t-1}[m].$$

    **else**

$$\mathbf{W}_t[m] \leftarrow (1 - \delta) \, \mathbf{W}_{t-1}[m] + \delta \, \mathbf{W}_t[m].$$

**return** $int(\mathbf{W}_t[m])$

---

---
**Algorithm 4 TopK**$(m, k, t)$
---
The local class contributions are sorted:

$$sorted\_list \leftarrow sort(\{|D_j|[m]\}_{j=1}^{J}).$$

The top-$k$ contributing clients are selected:

$$sorted_k \leftarrow sorted\_list[0 : k].$$

The masks from the selected clients are aggregated using logical OR:

$$\mathbf{S}_t[m] \leftarrow \bigvee_{j \in sorted_k} \mathbf{S}_t^j[m].$$

**return** $\mathbf{S}_t[m]$

---

### A.2.2 CS-pFedTM

---

**Algorithm 5 CS-pFedTM: Communication-Efficient and Similarity-Driven Personalisation with TM**

---

The total number of clients is denoted by $N_c$, the total number of communication rounds by $T$, the number of clauses per client by $n_{\text{clauses}}$, and the communication budget by $\tau$.

**for** round $t = 0, 1, \dots, T$ **do**

  A subset of clients $\mathcal{C}_t$ is sampled by the server.

  **if** $t = 0$ **then**

    A tiny reference TM is trained by all clients, and the resulting state parameters are uploaded to the server.

    `min_frac` is computed using **compute_min_frac**.

    Client similarity scores $JS_{\text{clients}}$ are computed using **compute_client_similarity**.

    The proportion of local clauses is determined:

$$\texttt{local\_frac} \leftarrow \exp\left(-\ln\left(\frac{1}{\texttt{min\_frac}}\right) \cdot JS_{\text{clients}}\right).$$

    Local and global clauses are assigned as:

$$n_{\text{local}} = \lfloor n_{\text{clauses}} \cdot \texttt{local\_frac} \rfloor, \qquad n_{\text{global}} = n_{\text{clauses}} - n_{\text{local}}.$$

  **for** each client $n \in \mathcal{C}_t$ **do**

    A local model $L^n$ and a global model $G^n$ are trained.

    Clause masks are applied:

$$L^n \leftarrow \textbf{mask\_weights}(L^n), \qquad G^n \leftarrow \textbf{mask\_weights}(G^n).$$

    The global parameters $G^n$ are uploaded to the server.

  A global aggregation step is performed:

$$G_t \leftarrow \textbf{aggregate\_global\_models}.$$

  The aggregated global TM $G_t$ is distributed to all participating clients.

  **return** Personalised TMs for each client, denoted by $TM^n \in \{G_t, L^n\}$, combined according to Equation 3.

---

**Algorithm 6 compute_min_frac($\tau, n_{clauses}$)**

---

`per_clause_size` $\leftarrow \frac{ref\_model\_size}{ref\_num\_clauses}$ `max_global_clauses` $\leftarrow \min\left(\lfloor\frac{\tau}{\texttt{per\_clause\_size}}\rfloor, \frac{n\_clauses}{2}\right)$ `min_local_clauses` $\leftarrow n_{clauses} -$ `max_global_clauses` Minimum local fraction: `min_frac` $\leftarrow \frac{\texttt{min\_local\_clauses}}{n_{clauses}}$ **return** `min_frac`

---

---

**Algorithm 7 compute_client_similarity**

---

The variable total_similarity is initialised to 0.

The variable pair_count is initialised to 0.

**for** each pair in combinations(len(*all_states*), 2) **do**

    The pairwise similarity is added:

$$\text{total\_similarity} \leftarrow \text{total\_similarity} + \textbf{JSTest}(pair[0], pair[1]).$$

    The pair count is incremented:

$$\text{pair\_count} \leftarrow \text{pair\_count} + 1.$$

    The average similarity is computed:

$$\text{average\_jaccard\_similarity} \leftarrow \begin{cases} \frac{\text{total\_similarity}}{\text{pair\_count}}, & \text{if pair\_count} > 0 \\ 0, & \text{otherwise} \end{cases}$$

**return** average_jaccard_similarity

---

**Algorithm 8 JSTest**$(S^A, S^B)$

---

**if** $\text{len}(S^A) \neq \text{len}(S^B)$ **then**

    A value error is raised indicating that both vectors must have the same length.

The intersection is computed:

$$\text{intersection} \leftarrow \sum_{i=0}^{\text{len}(S^A)} \left( S^A[i] \wedge S^B[i] \right).$$

The union is computed:

$$\text{union} \leftarrow \sum_{i=0}^{\text{len}(S^A)} \left( S^A[i] \vee S^B[i] \right).$$

**if** union $= 0$ **then**

    **return** 0

**else**

    **return** $\frac{\text{intersection}}{\text{union}}$

---

**Algorithm 9 mask_weights**$(W)$

---

**for** class $m = 1, 2, \ldots, M$ **do**

    **if** class $m$ is not present in the local dataset **then**

        The weight for class $m$ is set to zero:

$$W[m] \leftarrow 0.$$

**return** $W$

---

**Algorithm 10 aggregate_global_models**

---

A set of global models $\mathcal{G}_t$ is received, where each model $G^n \in \mathcal{G}_t$ contains the weights $W_t^n$ and states $S_t^n$.

Clients are ranked by their average confidence per class, denoted by $rank\_clients_t$.

Global weights $\mathbf{W}_t$ and states $\mathbf{S}_t$ are prepared for aggregation.

**for** class $m = 1, \ldots, M$ **do**

    $\mathbf{W}_t[m] \leftarrow \textbf{AverageCW}_{\textbf{CA}}(m, \delta, t)$

    $\mathbf{S}_t[m] \leftarrow \textbf{Top2}_{\textbf{Conf}}(m, rank\_clients_t, t)$

**return** $G_t = \{\mathbf{W}_t, \mathbf{S}_t\}$

---

---

**Algorithm 11 AverageCW$_{\textbf{CA}}(m, \delta, t)$**

---

A weighted average of local clause weights is computed:

$$\mathbf{W}_t[m] \leftarrow int\left(\frac{1}{|D|}\sum_{j=1}^{J}|D_j|\,\mathbf{W}_t^j[m]\right).$$

**if** $t > 1$ **then**

    **if** $\forall_{j=1}^{J}\,\mathbf{W}_t^j[m] = 0$ **then**

        The previous weights are retained when class $m$ is not observed:

$$\mathbf{W}_t[m] \leftarrow \mathbf{W}_{t-1}[m].$$

    **else**

$$\mathbf{W}_t[m] \leftarrow \delta\,\mathbf{W}_t[m].$$

**return** $\mathbf{W}_t[m]$

---

**Algorithm 12 Top2$_{\textbf{Conf}}(m, rank\_clients, t)$**

---

**if** $rank\_clients[m]$ contains at least two clients **then**

    The top two clients' states for class $m$ are combined using logical OR:

$$\mathbf{S}_t[m] \leftarrow \mathbf{S}_t^{rank\_clients[m][0]}[m] \vee \mathbf{S}_t^{rank\_clients[m][1]}[m].$$

**else**

    Only the best client's mask is used:

$$\mathbf{S}_t[m] \leftarrow \mathbf{S}_t^{rank\_clients[m][0]}[m].$$

**return** $\mathbf{S}_t[m]$

---

### A.3   Additional Results

### A.3.1   Ablation Studies

Further ablation studies were conducted for $\tau$, which specifies the maximum communication cost permitted per client per round and determines the minimum fraction of clauses that must remain local (`min_frac`). This parameter influences only the number of parameters that can be transmitted by each client. Experiments on CIFAR-10 with varying values of $\tau$, presented in Table 8, show that at lower heterogeneity levels ($Dir(0.1)$ and $Dir(0.05)$), increasing $\tau$, thereby allowing more global clause sharing, leads to higher accuracy, as clients with substantial distributional overlap benefit from increased access to global knowledge. In contrast, under extreme heterogeneity ($Dir(0.01)$ and $Dir(0.005)$), increasing $\tau$ results in only marginal changes in performance, as personalisation dominates and additional global clauses contribute limited value.

Table 8: Ablation Studies for $\tau$.

| $\tau$ | $Dir(0.1)$ | $Dir(0.05)$ | $Dir(0.01)$ | $Dir(0.005)$ |
|---|---|---|---|---|
| 0.005 | 79.61±1.58 | 86.92±1.18 | **96.83±1.15** | **98.47±1.32** |
| 0.01 | 79.30±1.35 | **87.07±0.95** | 96.73±1.02 | 98.43±1.38 |
| 0.03 | 79.41±1.46 | 87.02±1.41 | 96.59±1.13 | 98.36±0.55 |
| 0.05 | **79.98±1.48** | 86.97±1.23 | 96.48±0.62 | 97.94±0.98 |
| 0.1 | 79.83±1.28 | 86.81±1.16 | 96.71±1.24 | 98.16±0.84 |

### A.3.2   Impact of the Modified Global Aggregation Scheme

To evaluate the proposed aggregation scheme, **AverageCW$_{\mathbf{CA}}$**, Figure 3 compares its convergence behaviour with the standard **AverageCW**. Across all datasets and heterogeneity settings, **AverageCW$_{\mathbf{CA}}$** consistently achieves higher accuracy throughout training. The performance gap is modest on CIFAR-10 but becomes increasingly pronounced on CIFAR-100 and Tiny-ImageNet, indicating that **AverageCW**, which combines current and previous round parameters, becomes more susceptible to aggregation skew as the number of classes increases. This effect is further amplified under greater data heterogeneity, where $Dir(0.05)$ exhibits a larger performance gap than $Dir(0.1)$. Hence, scaling only the current-round parameters in **AverageCW$_{\mathbf{CA}}$** mitigates this skew and leads to more stable convergence.

To compare sample-based **TopK**, and confidence-based states aggregation, **TopK$_{\mathbf{Conf}}$**, experiments were conducted on CIFAR-10 under both cross-silo (10 clients) and cross-device (100 clients) settings, with full (1.0) and partial (0.3) participation.

As shown in Table 9, confidence-based **TopK$_{\mathbf{Conf}}$**, achieves higher mean accuracy across all settings. In particular, in cross-device scenarios with 100 clients, confidence-based selection yields more stable performance, as reflected by lower variance under full participation.

In contrast, sample-based **TopK** tends to repeatedly select clients with the largest local datasets. This effect is particularly evident in the cross-silo setting, where all clients participate each round but only a small subset with the highest sample counts contributes to aggregation. While this behaviour can reduce variance when the number of clients is small, it introduces a selection bias as the client pool grows. In the 100-client setting, this bias leads to the over-representation of the same subset of dominant clients, resulting in higher variance and less consistent performance.

Overall, these results suggest that **TopK$_{\mathbf{Conf}}$** provides a fairer and more informative states aggregation mechanism by prioritising model confidence rather than dataset size. This promotes more balanced aggregation, particularly in large-scale and highly heterogeneous FL scenarios.

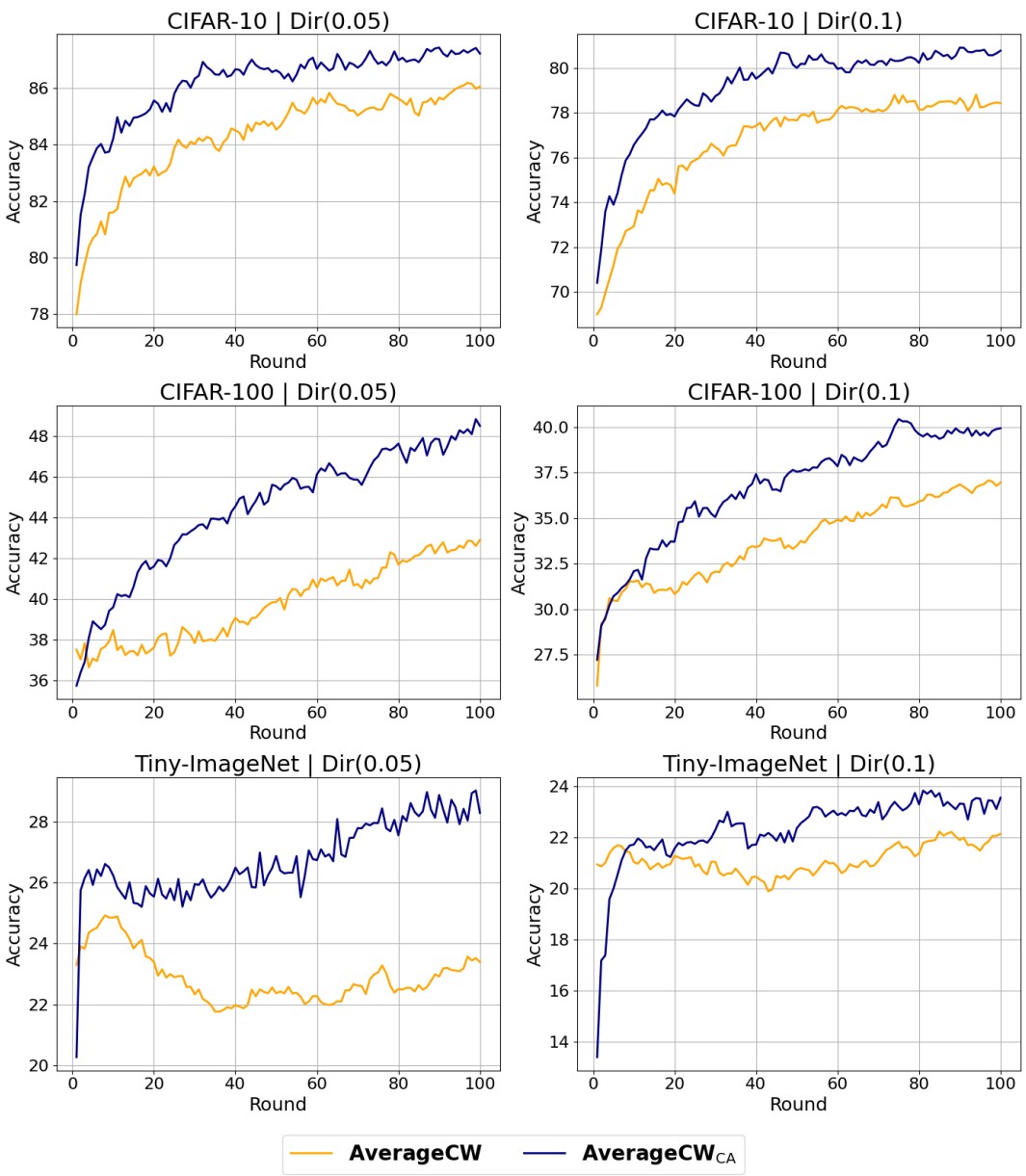

Figure 3: Comparison of **AverageCW** and **AverageCW**$_{CA}$ across CIFAR-10, CIFAR-100, and Tiny-ImageNet for $Dir(0.05)$ and $Dir(0.1)$.

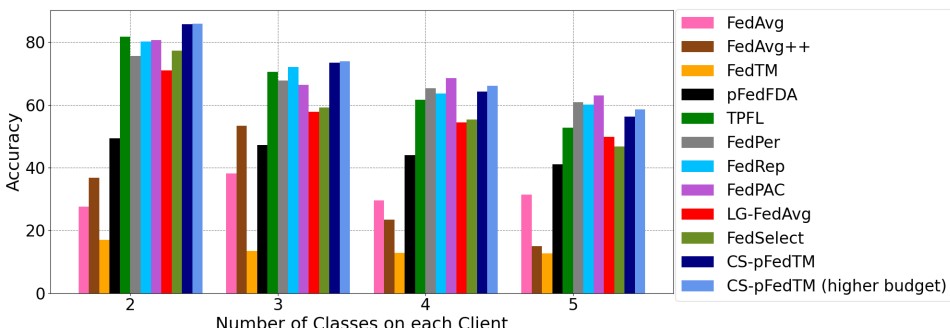

Figure 4: Performance of the algorithms on varying heterogeneity.

Table 9: Performance of CS-pFedTM with **TopK** and **TopK$_{\mathbf{Conf}}$** for various FL settings

| | | 10 Clients | 100 Clients | |
|---|---|---|---|---|
| | | Full Participation | Participation Ratio (0.3) | Full Participation |
| $Dir(0.05)$ | **TopK** | 84.53±1.81 | 86.83±0.99 | 87.33±1.99 |
| | **TopK$_{\mathbf{Conf}}$** | **84.75±2.13** | **86.92±1.18** | **87.73±0.69** |
| $Dir(0.1)$ | **TopK** | 76.67±1.71 | 79.17±0.63 | 79.89±1.82 |
| | **TopK$_{\mathbf{Conf}}$** | **79.85±2.30** | **79.61±1.58** | **80.88±0.49** |

### A.3.3 Impact of Label Skew on Performance

To analyse the impact of heterogeneity with label skew, the number of classes per client in CIFAR-10 was varied, with fewer classes indicating higher heterogeneity. As shown in Figure 4, CS-pFedTM achieves the largest gains under highly heterogeneous settings, although its advantage decreases slightly as heterogeneity is reduced. It consistently outperforms communication-efficient baselines such as LG-FedAvg, TPFL and FedSelect. Similar to CNN-based approaches, stronger performance under lower heterogeneity typically requires a larger proportion of shared global parameters, a trend that is also exhibited by CS-pFedTM.

For the higher-budget setting, CS-pFedTM was constrained to match the maximum communication cost used by competing methods; even under this constraint, it incurred substantially lower costs while narrowing the performance gap. Part of the remaining gap is attributed to the fact that TMs are generally less robust than CNNs (How et al., 2025), particularly as the number of local classes increases and the classification task becomes more complex, favouring the richer continuous representations learned by DNNs. However, CS-pFedTM remains the strongest TM-based FL method across all heterogeneity levels. Moreover, recent advances in TM architectures, such as GraphTM, indicate promising directions for further improvements in robustness and representational capacity (Granmo et al., 2025).

### A.3.4 Performance in Extreme Non-IID setting

Parameter similarity is expected to lose granularity under extreme non-IID conditions (eg. when clients have completely disjoint label spaces). In such cases, the similarity between client clause sets saturates near zero, leading the allocation mechanism to assign predominantly local clauses. This behaviour is desirable: when clients share almost no structural information, the model should naturally shift towards fully personalised learning.

To assess performance under these extreme regimes, CS-pFedTM was evaluated with Dirichlet settings of $\alpha = 0.01$ and $\alpha = 0.005$. As shown in Table 10, CS-pFedTM continues to match or exceed the performance of all baselines, even as the degree of global sharing diminishes. These results indicate that the similarity-driven allocation mechanism remains stable and effective, including in scenarios where the model transitions towards near-fully personalised operation.

Table 10: Performance of the algorithms in the extreme non-IID setting.

|  | $Dir(0.005)$ | $Dir(0.01)$ |
|---|---|---|
| FedAvg | 28.18±0.74 | 25.36±0.58 |
| FedAvg++ | 95.91±1.12 | 92.27±0.84 |
| pfedFDA | 94.89±0.66 | 91.86±0.91 |
| FedPAC | 97.81±0.93 | **96.98±0.77** |
| FedRep | 97.25±0.85 | 95.20±0.64 |
| FedPer | 98.04±1.02 | 96.67±0.72 |
| LG-FedAvg | **98.57±0.81** | 94.36±1.10 |
| FedSelect | 97.85±0.97 | 95.28±1.23 |
| TPFL | 98.38±1.37 | 96.29±1.01 |
| FedTM | 24.63±7.28 | 28.52±9.59 |
| CS-pFedTM | 98.47±1.32 | 96.83±1.15 |

### A.3.5 Scalability Analysis

The scalability of CS-pFedTM was assessed by varying the number of clients from 20 to 500 and adjusting the client participation ratio per communication round to $\{0.1, 0.3, 0.5, 1.0\}$ on the CIFAR-10 dataset. The results indicate that CS-pFedTM maintains strong and stable performance across all configurations, irrespective of the total number of clients or the participation rate. This demonstrates that the method scales effectively and remains robust across a wide range of FL configurations, supporting its practical applicability.

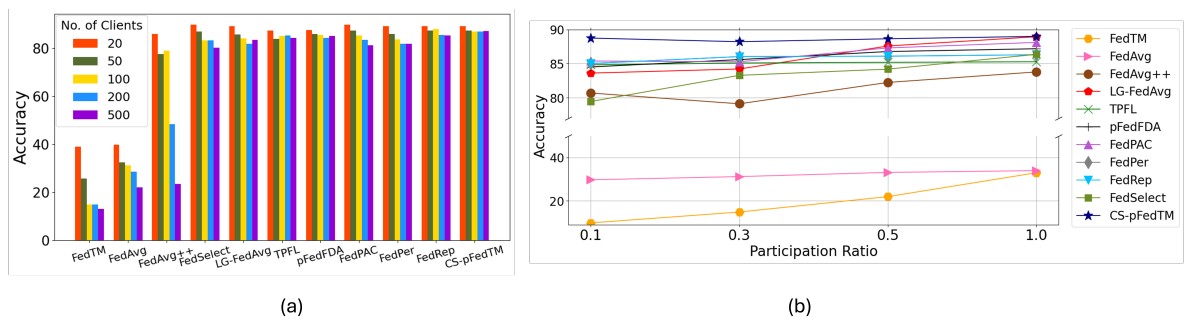

Figure 5: Performance of the algorithms for varying (a) number of clients and (b) participation ratio.

### A.3.6 Sensitivity Analysis

As shown in Figure 6, the similarity-driven allocation selects the optimal point on each curve by adaptively adjusting the local–global split according to the level of client heterogeneity. Upload communication costs increase as heterogeneity decreases, as a larger fraction of global clauses must be transmitted. Notably, the selected operating points fall in regions where the performance curves are relatively flat, indicating that small variations in the estimated similarity score, and hence small shifts in clause allocation, produce only marginal changes in accuracy. These results demonstrate the trade-off between personalisation and communication, indicating that the allocation mechanism identifies an effective balance across different heterogeneity settings.

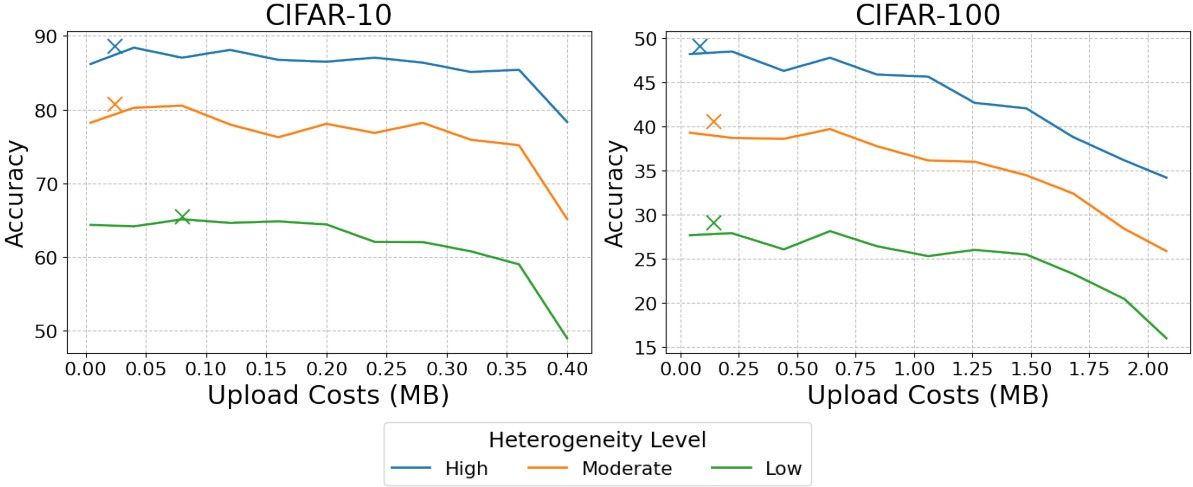

Figure 6: Performance as a function of the local clause fraction under different heterogeneity levels. 'x' indicates the local-global split selected by the similarity-driven allocation, which achieves the highest performance on each curve.

### A.3.7    Comparison with Sparsification Methods

Sparsification can also serve as a form of communication-efficient personalisation by selectively pruning model components according to their relevance for each client. To assess this perspective, CS-pFedTM is compared with DisPFL (Dai et al., 2022) and SpaFL (Kim et al., 2024), two communication-efficient personalised FL approaches that employ sparsification.

As DisPFL operates within a decentralised FL framework, the comparison is made using the average per-round communication cost per client when exchanging parameters with neighbouring clients. This is contrasted with the per-client communication cost incurred under CS-pFedTM. For fairness, the same CNN architectures described in Section 5 were used.

Although decentralised FL methods avoid the server-side communication bottleneck, they become increasingly communication-intensive when $n > 1$, since each client must exchange updates with multiple neighbours in every round. In contrast, centralised FL requires only a single upload and download per client. The results show that CS-pFedTM outperforms DisPFL across all evaluated settings while incurring substantially lower per-round communication costs. A notable advantage of DisPFL is that the number of neighbours can be predefined, providing flexibility in the design of the communication topology. However, this flexibility introduces a trade-off: as observed in Tables 11 increasing the number of neighbours may improve information mixing but also raises communication overhead and can adversely affect model performance.

To further reduce communication, pruning-based methods have been proposed. SpaFL assigns trainable thresholds to each filter or neuron so that the associated parameters can be pruned, inducing structured sparsity. Communication is reduced by transmitting only these thresholds between clients and the server, lowering costs by up to two orders of magnitude relative to exchanging full model parameters (Kim et al., 2024).

However, pruning is largely ineffective for smaller CNNs because their limited parameter counts offer little redundancy to exploit. As the CNNs used in Section 5 are too small for meaningful pruning, the larger model from the SpaFL paper (Kim et al., 2024) is adopted for comparison.

### A.3.8    Comparison with Larger Baseline Models

The primary focus of this work is efficiency-oriented personalised FL, in which comparisons are typically conducted under realistic communication and computational constraints. Under such conditions, lightweight

Table 11: Performance of DisPFL($n$), where $n$ denotes the number of neighbours, compared with CS-pFedTM, with the average communication cost (CC) per client per round.

| Dataset | Method | $Dir(0.05)$ | | $Dir(0.1)$ | |
|---------|--------|------|------|------|------|
| | | Acc | CC | Acc | CC |
| SVHN | DisPFL ($n$=30) | 77.08±2.17 | 6.46 | 65.09±2.24 | 6.46 |
| | DisPFL ($n$=10) | 75.96±1.93 | 2.15 | 60.18±1.92 | 2.15 |
| | DisPFL ($n$=5) | 76.35±2.01 | 1.08 | 61.36±1.85 | 1.08 |
| | CS-pFedTM | **89.59±0.95** | **0.006** | **83.67±0.68** | **0.006** |
| EMNIST | DisPFL ($n$=30) | 90.90±0.32 | 7.43 | 88.44±0.41 | 7.43 |
| | DisPFL ($n$=10) | 90.15±0.38 | 2.48 | 88.89±0.42 | 2.48 |
| | DisPFL ($n$=5) | 91.15±0.42 | 1.24 | 89.90±0.45 | 1.24 |
| | CS-pFedTM | **94.36±0.40** | **0.012** | **91.08±0.09** | **0.024** |
| CIFAR-10 | DisPFL ($n$=30) | 82.26±1.12 | 6.50 | 74.57±1.04 | 6.50 |
| | DisPFL ($n$=10) | 82.10±0.93 | 2.17 | 71.94±1.01 | 2.17 |
| | DisPFL ($n$=5) | 81.41±0.93 | 1.08 | 73.15±0.94 | 1.41 |
| | CS-pFedTM | **86.92±1.18** | **0.004** | **79.61±1.58** | **0.006** |
| CIFAR-100 | DisPFL ($n$=30) | 30.52±1.30 | 7.14 | 23.38±0.42 | 7.14 |
| | DisPFL ($n$=10) | 30.10±0.82 | 2.38 | 23.22±0.79 | 2.38 |
| | DisPFL ($n$=5) | 30.46±0.79 | 1.19 | 21.82±0.33 | 1.19 |
| | CS-pFedTM | **48.09±0.28** | **0.028** | **38.85±0.35** | **0.014** |
| Tiny-ImageNet | DisPFL ($n$=30) | 22.55±0.43 | 7.93 | 17.09±0.62 | 7.93 |
| | DisPFL ($n$=10) | 21.03±0.66 | 2.64 | 15.69±0.59 | 2.64 |
| | DisPFL ($n$=5) | 19.10±0.46 | 1.32 | 14.79±0.47 | 1.32 |
| | CS-pFedTM | **29.24±0.64** | **0.04** | **23.91±0.32** | **0.04** |

Table 12: Comparison of SpaFL and CS-pFedTM in performance, upload and download costs (CC), and model size after pruning.

| Dataset | Method | $Dir(0.05)$ | | $Dir(0.1)$ | | Size |
|---------|--------|------|------|------|------|------|
| | | Acc | CC | Acc | CC | |
| F-MNIST | SpaFL | 96.72±0.31 | 0.07/0.23 | 95.24±0.42 | 0.07/0.23 | 0.94 |
| | CS-pFedTM | **97.51±0.72** | **0.02/0.22** | **95.78±0.28** | **0.02/0.24** | **0.26** |
| CIFAR-10 | SpaFL | 83.33±0.79 | 0.09/0.26 | 75.57±0.65 | 0.09/0.26 | 3.23 |
| | CS-pFedTM | **86.92±1.18** | **0.004/0.09** | **79.61±1.58** | **0.006/0.13** | **0.09** |
| CIFAR-100 | SpaFL | 45.15±0.94 | 0.29/0.96 | 36.25±0.64 | 0.29/0.96 | 11.2 |
| | CS-pFedTM | **48.09±0.28** | **0.028/0.44** | **38.85±0.35** | **0.014/0.35** | **0.46** |

CNNs remain the standard choice in recent personalisation literature, as they more accurately reflect practical FL deployments. Nevertheless, the DNN-based FL methods were also evaluated using MobileNet-v2 Howard et al. (2019) on CIFAR-10.

The results in Table 13 align with the expected behaviour of parameter-decoupled FL methods: approaches that personalise a substantial proportion of the model (eg. LG-FedAvg) retain reasonable performance even when MobileNet is used, since a large number of parameters are adapted locally. In contrast, methods that personalise only the classifier head (FedPer, FedRep, FedPAC) perform worse than with the 2-layer CNN. This is anticipated, as MobileNet's large shared backbone dominates the learned representations, and personalising only the final layer is insufficient to compensate for strong distribution shifts under heterogeneous data.

Importantly, even with a significantly larger backbone, the communication costs of these MobileNet-based baselines remain several orders of magnitude higher than those of CS-pFedTM. Despite relying on a lightweight architecture, CS-pFedTM attains comparable accuracy while preserving its principal advantage of substantially reduced communication.

Table 13: Performance comparison of the algorithms with a larger baseline model.

| | $Dir(0.05)$ | | $Dir(0.1)$ | |
|---|---|---|---|---|
| | Acc | CC | Acc | CC |
| FedAvg | 39.73±1.99 | 268/895 | 33.25±1.34 | 268/895 |
| FedAvg++ | 72.40±1.31 | " | 60.53±1.02 | " |
| pFedFDA | 88.01±1.17 | " | 80.93±0.91 | " |
| FedPAC | 75.29±0.96 | 267/890 | 69.64±0.83 | 267/890 |
| FedRep | 80.72±1.25 | " | 78.44±1.11 | " |
| FedPer | 76.79±0.92 | " | 66.17±0.73 | " |
| LG-FedAvg | 87.68±1.83 | 6.66/10.2 | **83.21±1.55** | 6.66/10.2 |
| FedSelect | 85.35±0.93 | 243/813 | 79.89±0.85 | 243/813 |
| TPFL | 87.52±0.54 | 0.39/1.3 | 80.12±1.33 | 0.39/1.3 |
| CS-pFedTM | **88.02±0.71** | **0.04/0.88** | 82.74±0.83 | **0.04/0.88** |

## A.4 Further Discussion

**Stability of Parameter Similarity and Use of the Reference Round.** The reference round is employed solely to estimate the parameter similarity that determines clause allocation. Because clients are sampled uniformly at random in every round, including the reference round, the participating subset constitutes an unbiased sample of the overall population. Consequently, the similarity estimated in this initial round provides a reliable indicator of the underlying system heterogeneity.

Empirically, client parameter similarity was computed at every training round, and its variance across rounds was evaluated for different client participation rates (0.1, 0.3, 0.5, 1.0), averaged over three independent random seeds. As shown in Table 14, the variance remains extremely small across all datasets and heterogeneity levels, indicating that similarity is tightly concentrated around the value obtained during the reference round. Although the variance decreases marginally as the participation rate increases, the reduction is minor, demonstrating that similarity is already highly stable even under low participation. These findings confirm that the reference-round similarity estimate is reliable and remains robust irrespective of the sampling rate.

Table 14: Average variance of parameter similarity across training rounds.

| | Participation Ratio | SVHN | EMNIST | CIFAR-10 | CIFAR-100 | Tiny-Imagenet |
|---|---|---|---|---|---|---|
| $Dir(0.05)$ | 0.1 | 0.0120 | 0.0007 | 0.0054 | 0.0005 | 0.0006 |
| | 0.3 | 0.0047 | 0.0006 | 0.0028 | 0.0004 | 0.0008 |
| | 0.5 | 0.0029 | 0.0005 | 0.0023 | 0.0004 | 0.0005 |
| | 1 | 0.0021 | 0.0005 | 0.0023 | 0.0003 | 0.0005 |
| $Dir(0.1)$ | 0.1 | 0.0023 | 0.0040 | 0.0116 | 0.0011 | 0.0013 |
| | 0.3 | 0.0053 | 0.0028 | 0.0099 | 0.0009 | 0.0007 |
| | 0.5 | 0.0015 | 0.0021 | 0.0058 | 0.0009 | 0.0002 |
| | 1 | 0.0019 | 0.0011 | 0.0035 | 0.0009 | 0.0002 |

Moreover, Figure 5 shows that model performance remains stable across different participation rates (full participation versus partial participation). If the similarity estimate were highly sensitive to the specific clients participating in any given round, substantial divergence in accuracy across participation settings would be expected. Instead, accuracy remains nearly unchanged, indicating that the heterogeneity captured during the reference round is representative of the system in subsequent rounds. Consistently low variance in overall performance is also observed across repeated runs, reinforcing that system behaviour does not fluctuate meaningfully with changes in the sampled client set.

With regard to dynamic data distributions (concept drift), CS-pFedTM is naturally compatible with such scenarios. Since global parameters are already exchanged in every communication round, the system can periodically re-estimate inter-client similarity (for example, every $N$ rounds) and update clause allocation accordingly, without requiring modifications to the core algorithm or incurring additional communication cost.

**Limitations.** Although CS-pFedTM performs well under high heterogeneity, its effectiveness may diminish when clients receive too few global clauses to contribute meaningfully, whether due to device limitations or a stringent communication budget imposed by $\tau$. While $\tau$ functions exactly as intended, governing communication cost without altering the personalisation dynamics, very restrictive budgets can nonetheless limit the extent to which certain clients are able to exploit global structure. This observation motivates future work on adaptive clause allocation strategies that remain effective under severe device or communication constraints. Additionally, as heterogeneity decreases, the performance gap relative to DNN-based methods narrows, partly attributable to the inherent representational differences between TMs and DNNs under more complex multi-class settings, which is a known trade-off of adopting a lightweight logic-based architecture (How et al., 2025; Tarasyuk et al., 2026).

**Future Directions.** Addressing the challenges identified above gives rise to several promising avenues for advancing personalisation in TM-based FL. In particular, the exploration of intra-clause weight optimisation, adaptive or learned personalised masks, and clause-level sparsification or pruning may enhance both efficiency and personalisation. Furthermore, the observed relationship between parameter similarity and distributional divergence suggests broader opportunities for state-based similarity measures, including dynamic clause allocation and resource-aware personalisation tailored to heterogeneous device capabilities. Extending the evaluation to feature skew and mixed heterogeneity scenarios, potentially leveraging multi-domain benchmarks, is another promising avenue as TM architectures continue to mature.

While CS-pFedTM currently uses separate global and local clause sets, a simpler formulation could maintain a unified clause pool, dynamically designating clauses as shared or personalised based on continuously updated similarity estimates, which would be particularly valuable in settings with evolving client distributions. Furthermore, client-specific allocation, where each client's local-global split is determined by its individual heterogeneity relative to the system rather than a single system-level estimate, remains an interesting open direction.

