# OpenReview forum: "CS-pFedTM: Communication-Efficient and Similarity-based Personalised Federated Learning with Tsetlin Machine"
_TMLR — Under review for TMLR_

### Review · Reviewer_zoaX · 2026-03-14

**Summary Of Contributions:**

This paper proposes a Tsetlin Machine (TM) based framework for personalized federated learning where each client trains a local and global TM which is synchronized with a global server. The paper makes a number of design decisions to optimize communication and other objectives for this setting by measuring data heterogeneity and customizing the aggregation scheme. The method is then applied to a number of tiny models on toy-scale datasets, showing large efficiency gains of what appears to be generic federated learning algorithms applied to TMs, as well as FedTM.

Strengths:
- The paper and experiments seem to be largely complete. Since I lack understanding of the standards and literature in this area particular area, I cannot comment further.
- The claimed efficiency gains are substantial.

Weaknesses:
- The only evaluations seem to be on tiny models on toy datasets. By the standards of areas which I am familiar with, I would broadly consider the evaluations to be unconvincing due to scale, with several orders of magnitude larger scale required to show credible results, though I understand this standard is not generally applied in this area.
- I'm highly skeptical that this problem actually exists. The problems described are fairly concrete and extremely specific, and possibly vacuous.

**Audience:**

No

**Audience Explanation:**

I'm not sure what this paper is for, and it seems to combine a bunch of things which are often written abstractly in a way that implies them to be useful, but without a tangible example of what they can be used for. Overall, the topic becomes so specific and off-mainstream that the niche that it occupies is probably empty, as I cannot imagine any practitioner ever using this technique or anything based on this.

- I'm not sure what the point of a Tsetlin Machine is. To my knowledge, there are no real applications of this yet, so trying to make what is effectively a third order extension (any Tsetlin Machine -> TM on Edge Devices -> Federated Learning of TM -> Personalized FL on TM...) seems completely premature. The statement that smaller models are needed and sufficient also runs completely contrary to the general trend.
- I'm also highly skeptical of personalized FL in general. Again, this runs completely contrary to the general trends we see in ML towards centralization. The only place where FL has really caught on in my understanding is cross-silo applications where data is siloed due to regulatory reasons, while the research proposed here seems to go in the complete opposite direction of many small clients.

**Broader Impact Concerns:**

No impact concerns.

**Claims And Evidence:**

Yes

**Claims Explanation:**

Due to a complete lack of knowledge about Tsetlin Machines and only limited background in federated learning, I lack the expertise to properly evaluate the technical claims provided in the paper against the standards of the field. Overall, it seems fine and thorough, and my understanding is that toy-scale datasets (e.g., CIFAR, MNIST) with small numbers of clients and naive, simulated heterogeneity (i.e., class-based but otherwise IID) are commonly used in FL.

**Requested Changes:**

I don't consider myself qualified to comment on the technical correctness of this work, so will defer to other reviewers.

It would be good for the authors and/or other reviewers who are more familiar with this work to explain why this is needed and who this is for. Any concrete examples of how this could be used (i.e., where it would not be easily and clearly dominated by a conventional solution) would be helpful. If other reviewers are convinced that this is somehow useful I will defer to them.

---

### Review · Reviewer_gYBZ · 2026-04-23

**Summary Of Contributions:**

This paper proposes a personalized federated learning method CS-pFedTM based on Tsetlin Machines. The method combines local/global clause splitting, similarity-based adaptive allocation, class-specific masking, and confidence-based aggregation to improve communication efficiency and deal with data heterogeneity across clients.

**Audience:**

Yes

**Audience Explanation:**

Federated learning allows clients to train models locally and only share model parameters, but it still faces two main challenges: data heterogeneity and communication limits. This paper aims to address both issues.

**Claims And Evidence:**

Yes

**Claims Explanation:**

The experiments on several datasets show that the method can reach accuracy close to strong personalized FL baselines, while greatly reducing communication and runtime cost.

**Requested Changes:**

- The paper introduces an inverse relationship between Wasserstein distance and Jaccard similarity, and uses it to support the adaptive allocation strategy. However, this part currently feels more intuitive than fully convincing.
- CS-pFedTM includes several components, such as similarity-based clause allocation, communication-aware budgeting, class-specific masking, TopKConf, and AverageCWCA. Because of this, the paper should include ablation results to show how much each part contributes.
- Figure 1 shows the overall relationship between the local fraction and model performance, but the paper should further study how sensitive the method is to small changes in the chosen allocation, and whether the same pattern holds across different datasets and heterogeneity levels.
- Since the method depends heavily on parameter similarity as an indicator of data heterogeneity, the paper should provide stronger and more direct evidence that this assumption is valid.
- The experiments mainly consider Dirichlet heterogeneity with $\alpha \in \{0.1, 0.05\}$. It would be better to also test other heterogeneous settings, such as milder non-IID cases, label skew, feature skew, or mixed scenarios.
- The paper should also report results under different communication budgets. Since communication-aware allocation is one of the main claims, it would be useful to see how performance changes when the budget becomes tighter or more relaxed.
- The method needs an initial reference round to estimate communication cost per clause and client similarity. The paper should report the cost of this step more clearly.

---

### Review · Reviewer_bHES · 2026-06-22

**Summary Of Contributions:**

Considering communication efficiency, client heterogeneity, and client model size, this paper proposed methods on TM-based personalized FL with similarity-driven, budget-aware clause allocation. The efficiency gains (communication, runtime memory, latency) are large and convincing, with broad experiments (ablations, scalability, extreme non-IID, sparsification and larger-model comparisons).

The main concerns focus on those aspects:

 1. **Mismatch between the client-level motivation and the system-wide allocation actually implemented.** The method is motivated by the claim that more-heterogeneous clients should keep more clauses local while less-heterogeneous clients rely more on global clauses, yet the mechanism computes a single average similarity J(clients) and derives one scalar local_frac, yielding the same n_local/n_global for every client (Alg. 1) The authors should either reframe the contribution as system-level heterogeneity-adaptive allocation, or make the allocation client-specific: e.g., using each client's average similarity to the rest of the federation.

 2. **Too few runs and large variance.** Results are averaged over only three runs and several standard deviations are very large (e.g., FedAvg SVHN 29.16±14.42, FedTM CIFAR-10 13.93±5.10). The authors should increase the number of runs and report confidence intervals so the true effect can be assessed.

3. **The Wasserstein–Jaccard relation is a motivating intuition, not a guarantee.** It is stated as Corollary 1 (restated identically as Corollary 3), but the proof in App. A. 1 only bounds the expected intersection and never models the Jaccard denominator (the union), so "intersection maximized" does not imply "ratio maximized"; it also relies on an unbounded approximation $\operatorname{Pr}(\mathrm{L} \geq 1) \approx \mathrm{E}[\mathrm{L}] / \mathrm{L} $ max and an AM-GM step that establish an extremum only at $\mathrm{W}=0$, not the claimed monotonic implication. The argument therefore reads as intuition supported by the empirical fit in Fig. 2 rather than a rigorous result. The authors should either downgrade it to an assumption/empirical observation, or supply explicit assumptions and a probabilistic analysis over the full ratio.

4. **Put ablation in main part.** The method combines several components,  but the main tables report only end-to-end numbers. At least a concise ablation (similarity allocation, masking, TopKConf, AverageCWCA, budget-aware allocation) plus the equal-clause TPFL comparison should appear in the main text.

**Additional Comments:**

Currently the model used in this paper is kind of too small: e.g., CNN only contains two layers while the size of TM is smaller than 1MB.

I am not sure if it's the standard setting for TM-FL community but I think for modern mobile device, larger model is already applicable to deploy to. So I am curious about the result when both CNN and TM are larger to see if TM still keeps its efficiency on training and inferencing.

**Audience:**

Yes

**Audience Explanation:**

Communication efficiency is one of most principle factor that limit the applicability of FL method. This paper explored TM-based FL which focuses on deploying small models (compared to Neural Network type methods) to edge/mobile device, which might be of interest to the edge community.

**Broader Impact Concerns:**

No ethical concern

**Claims And Evidence:**

Yes

**Claims Explanation:**

In general, the experiments are concrete and complete. The revision I suggested has been put above.

**Requested Changes:**

The major issues have been listed in summary above.

Minor issues:

1. Check the consistency of the number used in paper. E.g., download 206×（§5.2）vs 210×（Abstract/Conclusion).

2. Corollary 1 is the same as Corollary 3. Might delete one of them.

3. Kindly improve the writing to make it more reader-friendly.